# Scalable Approximate Message Passing for Bayesian Neural Networks

## Abstract

Bayesian neural networks (BNNs) offer the potential for reliable uncertainty quantification and interpretability, which are critical for trustworthy AI in high-stakes domains. However, existing methods often struggle with issues such as overconfidence, hyperparameter sensitivity, and posterior collapse, leaving room for alternative approaches. In this work, we advance message passing (MP) for BNNs and present a novel framework that models the predictive posterior as a factor graph. To the best of our knowledge, our framework is the first MP method that handles convolutional neural networks and avoids double-counting training data, a limitation of previous MP methods that causes overconfidence. We evaluate our approach on CIFAR-10 with a convolutional neural network of roughly 890k parameters and find that it can compete with the SOTA baselines AdamW and IVON, even having an edge in terms of calibration. On synthetic data, we validate the uncertainty estimates and observe a strong correlation (0.9) between posterior credible intervals and its probability of covering the true data-generating function outside the training range. While our method scales to an MLP with 5.6 million parameters, further improvements are necessary to match the scale and performance of state-of-the-art variational inference methods.

## 1 Introduction

Deep learning models have achieved impressive results across various domains, including natural language processing (Vaswani et al., 2023), computer vision (Ravi et al., 2024), and autonomous systems (Bojarski et al., 2016). Yet, they often produce overconfident but incorrect predictions, particularly in ambiguous or out-of-distribution scenarios. Without the ability to effectively quantify uncertainty, this can foster both overreliance and underreliance on models, as users stop trusting their outputs entirely (Zhang et al., 2024), and in high-stakes domains like healthcare or autonomous driving, its application can be dangerous (Henne et al., 2020). To ensure safer deployment in these settings, models must not only predict outcomes but also express how uncertain they are about those predictions to allow for informed decision-making.

Bayesian neural networks (BNNs) offer a principled way to quantify uncertainty by capturing a posterior distribution over the model's weights, rather than relying on point estimates as in traditional neural networks. This allows BNNs to express epistemic uncertainty, the model's lack of knowledge about the underlying data distribution. Current methods for posterior approximation largely fall into two categories: sampling-based methods, such as Hamiltonian Monte Carlo (HMC), and deterministic approaches like variational inference (VI). While sampling methods are usually computationally expensive, VI has become increasingly scalable (Shen et al., 2024). However, VI is not without limitations: It often struggles with overconfidence (Papamarkou et al., 2024), and it can struggle to break symmetry when multiple modes are close (Zhang et al., 2018). Mean-field approaches, commonly used in VI, are prone to posterior collapse (Kurle et al., 2022; Coker et al., 2022). Additionally, VI often requires complex hyperparameter tuning (Osawa et al., 2019), which complicates its practical deployment in real-world settings. These challenges motivate the need for alternative approaches that can potentially address some of the shortcomings of VI while maintaining its scalability.

In contrast, message passing (MP) (Minka, 2001) is a probabilistic inference technique that suffers less from these problems. Belief propagation (Kschischang et al., 2001), the basis for many MP algorithms, integrates over variables of a joint density $p(x_1, \ldots, x_n)$ that factorize into a product of

functions $f_j$ on subsets of random variables $x_1, \ldots, x_n$. The corresponding factor graph is bipartite and connects these factors $f_j$ with the variables they depend on. The following recursive equations yield a computationally efficient algorithm to compute all marginals $p(x_i)$ for acyclic factor graphs:

$$p(x) = \prod_{f \in N_x} m_{f \to x}(x) \qquad m_{f \to x}(x) = \int f(N_f) \prod_{y \in N_f \setminus \{x\}} m_{y \to f}(y) \, d(N_f \setminus \{x\})$$

where $N_v$ denotes the neighborhood of vertex $v$ and $m_{y \to f}(y) = \prod_{f' \in N_y \setminus \{f\}} m_{f' \to y}(y)$. Since exact messages are often intractable and factor graphs are rarely acyclic, belief propagation typically cannot be applied directly. Instead, messages $m_{f \to X}(\cdot)$ and marginals $p_X(\cdot)$ are typically approximated by some family of distributions that has few parameters (e.g., Gaussians). However, applying message passing (MP) in practice presents two main challenges for practitioners: the need to derive (approximate) message equations when $m_{f \to x}$ falls outside the approximating family, and the complexity of implementing MP compared to other methods.

We summarize our contributions as follows:

1. We propose a scalable message-passing framework for Bayesian neural networks and derive message equations for various factors, which can benefit factor graph modeling across domains.

2. We implement our method in Julia for both CPU and GPU, and demonstrate its scalability to convolutional neural networks (CNNs) and large multilayer perceptrons (MLPs).

3. We evaluate on CIFAR-10 and find that our method is competitive with the SOTA baselines AdamW and IVON, even having an edge in terms of calibration while requiring no hyperparameter tuning.

To the best of our knowledge, this is the first MP method to handle CNNs and to avoid double-counting training data, thereby preventing overconfidence and, eventually, posterior collapse. While our methods scales to an MLP with 5.6 million parameters, further refinements are necessary to match the scale and performance of state-of-the-art VI methods.

## 1.1 RELATED WORK

As the exact posterior is intractable for most practical neural networks, approximate methods are essential for scalable BNNs. These methods generally fall into two categories: sampling-based approaches and those that approximate the posterior with parameterized distributions.

**Markov Chain Monte Carlo** (MCMC) methods attempt to draw representative samples from posterior distributions. Although methods such as Hamiltonian Monte Carlo are asymptotically exact, they become computationally prohibitive for large neural networks due to their high-dimensional parameter spaces and complex energy landscapes (Coker et al., 2022). An adaptation of Gibbs sampling has been scaled to MNIST, but on a very small network with only 8,180 parameters (Papamarkou, 2023). Approximate sampling methods can be faster but still require a large number of samples, which complicates both training and inference. Although approaches like knowledge distillation (Korattikara et al., 2015) attempt to speed up inference, MCMC remains generally too inefficient for large-scale deep learning applications (Khan & Rue, 2024).

**Variational Inference** (VI) aims to approximate the intractable posterior distribution $p(\theta \mid \mathcal{D})$ by a variational posterior $q(\theta)$. The parameters of $q$ are optimized using gradients with respect to an objective function, which is typically a generalized form of the reverse KL divergence $D_{\mathrm{KL}}\left[q(\theta) \| p(\theta \mid \mathcal{D})\right]$. Early methods like (Graves, 2011) and Bayes By Backprop (Blundell et al., 2015) laid the foundation for applying VI to neural networks, but suffer from slow convergence and severe underfitting, especially for large models or small dataset sizes (Osawa et al., 2019). More recently, VOGN (Osawa et al., 2019) achieved Adam-like results on ImageNet LSVRC by applying a Gauss-Newton approximation to the Hessian matrix. IVON (Shen et al., 2024) improved upon VOGN by using cheaper Hessian approximations and training techniques like gradient clipping, achieving Adam-like performance on large-scale models such as GPT-2 while maintaining similar runtime costs. Despite recent advancements, VI continues to face challenges such as overconfidence, posterior collapse, and complex hyperparameter tuning (see introduction), motivating the exploration of alternative approaches (Zhang et al., 2018).

**Message Passing for Neural Networks**: Message passing is a general framework that unifies several algorithms (Kschischang et al., 2001; Minka, 2001), but its direct application to neural networks

has been limited. Expectation backpropagation (EBP) (Soudry et al., 2014) approximates the posterior of 3-layer MLPs by combining expectation propagation, an approximate message passing algorithm, with gradient backpropagation. Similarly, probabilistic backpropagation (PBP) (Hernández-Lobato & Adams, 2015) combines belief propagation with gradient backpropagation and was found to produce better approximations than EBP (Ghosh et al., 2016). However, PBP treats the data as new examples in each consecutive epoch (double-counting), which makes it prone to overconfidence. Furthermore, EBP and PBP were both only deployed on small datasets and rely on gradients instead of pure message passing. In contrast, Lucibello et al. (2022) applied message passing to larger architectures by modeling the posterior over neural network weights as a factor graph, but faced posterior collapse to a point measure due to also double-counting data. Their experiments were mostly restricted to three-layer MLPs without biases and with binary weights. Our approach builds on this by introducing a message-passing framework for BNNs that avoids double-counting, scales to CNNs, and effectively supports continuous weights.

## 2 THEORETICAL MODEL

Our goal is to model the predictive posterior of a BNN as a factor graph and find a Gaussian approximation of the predictive posterior via belief propagation. Essentially, factor graphs are probabilistic modelling tools for approximating the marginals of joint distributions, provided that they factorize sufficiently. For a more comprehensive introduction on factor graphs and the sum-product algorithm, refer to Kschischang et al. (2001) BNNs, on the other hand, treat the parameters $\theta$ of a model $f_\theta : \mathbb{R}^d \longrightarrow \mathbb{R}^o$ as random variables with prior beliefs $p(\theta)$. Given a training dataset $\mathcal{D} = \{\boldsymbol{x}_i, \boldsymbol{y}_i\}_{i=1}^n$ of i.i.d. samples, a likelihood relationship $p(\boldsymbol{y} \,|\, \boldsymbol{x}, \theta) = p(\boldsymbol{y} \,|\, f_\theta(\boldsymbol{x}))$, and a new input sample $\boldsymbol{x}$, the goal is to approximate the predictive posterior distribution $p(\boldsymbol{y} \,|\, \boldsymbol{x}, \mathcal{D})$, which can be written as:

$$p(\boldsymbol{y} \,|\, \boldsymbol{x}, \mathcal{D}) = \int p(\boldsymbol{y} \,|\, \boldsymbol{x}, \theta) \, p(\theta \,|\, \mathcal{D}) \, d\theta. \tag{1}$$

This means that the density of the predictive posterior is the expected likelihood under the posterior distribution $p(\theta \,|\, \mathcal{D})$, which is proportional[1] to the product of the prior and dataset likelihood:

$$p(\theta \,|\, \mathcal{D}) \propto p(\theta) \prod_{i=1}^n p(\boldsymbol{y}_i \,|\, f_\theta(\boldsymbol{x})). \tag{2}$$

The integrand in Equation (1) exhibits a factorized structure that is well-suited to factor graph modeling. However, directly modelling the relationship $\boldsymbol{o} = f_\theta(\boldsymbol{x})$ with a single Dirac delta factor $\delta(\boldsymbol{o} - f_\theta(\boldsymbol{x}))$ does not yield feasible message equations. Therefore we model the neural network at scalar level by introducing intermediate latent variables connected by elementary Dirac delta factors. Figure 1 illustrates this construction for a simple MLP with independent weight matrices a priori. While the abstract factor graph in the figure uses vector variables for simplicity, we actually derive message equations where each vector component is treated as a separate scalar variable, and all Dirac deltas only depend on scalar variables. For instance, if $d = 2$, the conceptual factor $\delta(\mathbf{o} - \mathbf{W}_2\mathbf{a})$ is replaced by four scalar factors: $\delta(\mathrm{p}_{jk} - \mathrm{w}_{jk}\mathrm{a}_k)$ for $j, k = 1, 2$, with intermediate variables $\mathrm{p}_{jk}$, and two factors $\delta(\mathrm{o}_j - (\mathrm{p}_{j1} + \mathrm{p}_{j2}))$. By multiplying all factors in this expanded factor graph and integrating over intermediate results, we obtain a function in $\boldsymbol{x}, \boldsymbol{y}, \theta$ that is proportional to the integrand in Equation (1). Hence, the marginal of the unobserved target $\mathbf{y}$ is proportional to $p(\boldsymbol{y} \,|\, \boldsymbol{x}, \mathcal{D})$. When $\mathbf{y}$ connects to only one factor, its marginal matches its incoming message.

## 3 APPROXIMATIONS

Calculating a precise representation of the message to the target of an unseen input is intractable for large networks and datasets. The three primary reasons are, that a) nonlinearities and multiplication produce highly complex exact messages which are difficult to represent and propagate, b) the enormous size of the factor graph for large datasets, and c) the presence of various cycles in the graph. These challenges shape the message approximations as well as the design of our training and prediction procedures, which we address in the following sections.

---

[1]with a proportionality constant of $1/p(\mathcal{D})$

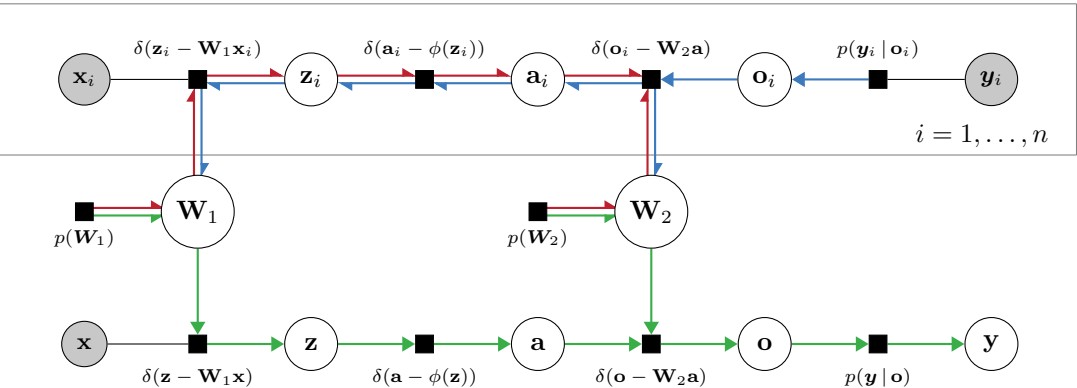

Figure 1: Conceptual vector-valued factor graph for a simple MLP. Each training example has its own "branch" (a copy of the network), while predictions for an unlabeled input $x$ are computed on a separate prediction branch. All branches are connected by the shared model parameters. Grayed-out variables are conditioned on (observed). Colored arrows indicate the three iteration orders: a forward / backward pass on training examples, and a forward pass for prediction.

### 3.1 APPROXIMATING MESSAGES VIA GAUSSIAN DENSITIES

To work around the highly complex exact messages, we approximate them with a parameterized class of functions. We desire this class to be closed under pointwise multiplication, as variable-to-factor messages are the product of incoming messages from other factors. We choose positive scalar multiples of one-dimensional Gaussian densities as our approximating family. Their closedness follows immediately from the exponential function's characteristic identity $\exp(x)\exp(y) = \exp(x + y)$ and the observation that for any $s_1, s_2 > 0$ and $\mu_1, \mu_2 \in \mathbb{R}$, the function $s_1(x-\mu_1)^2 + s_2(x-\mu_2)^2$ in $x$ can be represented as $s(x - \mu)^2 + c$ for some $s > 0$ and $\mu, c \in \mathbb{R}$. The precise relation between two scaled Gaussian densities and its product can be neatly expressed with the help of the so-called natural (re-)parameterization. Given a Gaussian $\mathcal{N}(\mu, \sigma^2)$, we call $\rho = 1/\sigma^2$ the precision and $\tau = \mu/\sigma^2$ the precision-mean. Collectively, $(\tau, \rho)$ are the Gaussian's natural parameters, $\mathbb{G}(x; \tau, \rho) := \mathcal{N}(x; \mu, \sigma^2)$, $x \in \mathbb{R}$. For $\mu_1, \mu_2 \in \mathbb{R}$ and $\sigma_1, \sigma_2 > 0$ with corresponding natural parameters $\rho_i = 1/\sigma_i^2$ and $\tau_i = \mu_i\rho_i$, $i = 1, 2$, multiplying Gaussian densities simplifies to:

$$\mathbb{G}(x; \tau_1, \rho_1) \cdot \mathbb{G}(x; \tau_2, \rho_2) = \mathcal{N}(\mu_1; \mu_2, \sigma_1^2 + \sigma_2^2) \cdot \mathbb{G}(x; \tau_1 + \tau_2, \rho_1 + \rho_2) \quad (3)$$

for all $x \in \mathbb{R}$. In other words, multiplying Gaussian densities simplifies to the pointwise addition of their natural parameters, aside from a multiplicative constant. Since we are only interested in the marginals, which are re-normalized, this constant does not affect the final result. Therefore, we can safely ignore these multiplicative constants and only keep track of the Gaussian's parameters.

Now we present our message approximations for three factor types, each representing a deterministic relationship between variables: ① the sum of variables weighted by constants, ② the application of a nonlinearity, and ③ the multiplication of two variables. As we model the factor graph on a scalar level, these three factors suffice to model complex modern network architectures such as ConvNeXt Liu et al. (2022)[2]. In E, we provide a comprehensive table of message equations, including additional factors for modeling training labels.

**Weighted Sum**: The density transformation property of the Dirac delta allows us to compute the exact message without approximation. For the relationship $s = c^\intercal v$ modeled by the factor $f := \delta(s - c^\intercal v)$, the message

$$m_{f \to s}(s) = \int \delta(s - c^\intercal v) \prod_{i=1}^{k} m_{v_i \to f}(v_i) \, dv_1 \ldots v_k$$

---

[2]with the exception of layer normalization, which can be substituted by orthogonal initialization schemes Xiao et al. (2018) or specific hyperparameters of a corresponding normalized network Nguyen et al. (2023)

is simply the density of $c^\intercal \mathbf{v}$, where $\mathbf{v} \sim \prod_{i=1}^{k} m_{v_i \to f}(v_i)$. If $m_{v_i \to f}(v_i) = \mathcal{N}(v_i; \mu_i, \sigma_i^2)$ are Gaussian, then $\mathbf{v} \sim \mathcal{N}(\boldsymbol{\mu}, \mathrm{diag}(\boldsymbol{\sigma}^2))$ and $m_{f \to s}(s)$ becomes a scaled multivariate Gaussian:

$$m_{f \to s}(s) = \mathcal{N}(s; c^\intercal \boldsymbol{\mu}, (c^2)^\intercal \boldsymbol{\sigma}^2).$$

The backward messages $m_{f \to v_i}$ can be derived similarly without approximation.

**Nonlinearity**: We model the application of a nonlinearity $\phi : \mathbb{R} \to \mathbb{R}$ as a factor $f := \delta(a - \phi(z))$. However, the forward and backward messages are problematic and require approximation–even for well-behaved, injective $\phi$ such as $\mathrm{LeakyReLU}_\alpha$:

$$m_{a \to f}(a) = \mathrm{pdf}_{\phi(Z)}(a) \ \text{ for } Z \sim \mathcal{N}$$

$$m_{f \to z}(z) = \int \delta(a - \phi(z)) \cdot m_{a \to f}(a) \, da = m_{a \to f}(\phi(z)) = \mathcal{N}(\phi(z); \mu_a, \sigma_a^2).$$

For values of $\alpha \neq 1$, the forward message is non-Gaussian and the backward message does not even integrate to 1. For ReLU ($\alpha = 0$), it is clearly not even integrable. Instead, we use *moment matching* to fit a Gaussian approximation. Given any factor $f$ and variable v, we can approximate the message $m_{f \to v}$ with a Gaussian if the moments $m_k := \int v^k m_{f \to v}(v) \, dv$ exist for $k = 0, 1, 2$ and can be computed efficiently:

$$m_{f \to v}(v) = \mathcal{N}(v; m_1/m_0, m_2/m_0 - (m_1/m_0)^2) \qquad \text{(Direct approximation)}$$

However, direct moment matching of the message is impossible for non-integrable messages or when the $m_k$ are expensive to find. Instead, we can apply moment matching to the updated *marginal* of v. Let $m_0, m_1, m_2$ be the moments of the "true" marginal

$$m(v) = \int f(v, v_1, ..., v_k) \ dv_1...dv_k \ \cdot \ \prod_i m_{g_i \to v}(v),$$

which is the product of the true message from $f$ and the approximated messages from other factors $g_i$. Then we can approximate $m$ with a Gaussian and obtain a message approximation

$$m_{f \to v}(v) := \mathcal{N}(v; \mu_v, \sigma_v^2)/m_{v \to f}(v) \qquad \text{(Marginal approximation)}$$

which approximates $m_{f \to v}$ so that it changes v's marginal in the same way as the actual message.[3] Since $m_{v \to f}(v)$ is a Gaussian density, we can compute $m_{f \to v}(v)$ efficiently by applying Gaussian division in natural parameters, similar to Equation (3).

For $\mathrm{LeakyReLU}_\alpha$, we found efficient direct and marginal approximations that are each applicable to both the forward and backward message when $\alpha \neq 0$. The marginal approximation remains applicable even for the ReLU case of $\alpha = 0$. We provide detailed derivations in Appendix B.2.

**Product** For the relationship c = ab, we employ variational message passing as in Stern et al. (2009), in order to break the vast number of symmetries in the true posterior of a Bayesian neural network. By combining the variational message equations for scalar products with the weighted sum, we can also construct efficient higher-order multiplication factors such as inner vector products. Refer to E for detailed equations.

### 3.2 TRAINING PROCEDURE & PREDICTION

In pure belief propagation, the product of incoming messages for any variable equals its marginal under the true posterior. With our aforementioned approximations, we can reasonably expect to converge on a diagonal Gaussian $\check{q}$ that approximates one of the various permutation modes of the true posterior by aligning the first two moments of the marginal. This concept can be elegantly interpreted through the lens of relative entropy. As shown in A.2, among diagonal Gaussians $q(\theta) = q_1(\theta_1) \cdots q_k(\theta_k)$, the relative entropy from (a mode of) the true posterior to $q$ is minimized for $\check{q}$:

$$\check{q} = \underset{q}{\mathrm{argmin}} \, D_{\mathrm{KL}}\left[\, p(\theta \,|\, \mathcal{D}) \,|\, q(\theta)\,\right]. \tag{4}$$

Another challenge in finding $\check{q}$ arises from cyclic dependencies. In acyclic factor graphs, each message depends only on previous messages from its subtree, allowing for exact propagation. However,

---

[3]This is the central idea behind expectation propagation as defined in Minka (2001).

our factor graph contains several cycles due to two primary reasons: ① multiple training branches interacting with shared parameters across linear layers, and ② the scalar-level modeling of matrix-vector multiplication in architectures with more than one hidden layer. These loops create circular dependencies among messages. To address these challenges, we adopt loopy belief propagation, where belief propagation is performed iteratively until messages converge. While exact propagation works in acyclic graphs, convergence is then only guaranteed under certain conditions (e.g., Simon's condition (Ihler et al., 2005)) that are difficult to verify. Instead, we pass messages in an iteration order that largely avoids loops by alternating forward and backward passes similarly to deterministic neural networks. Our message schedule is visualized in Figure 1.

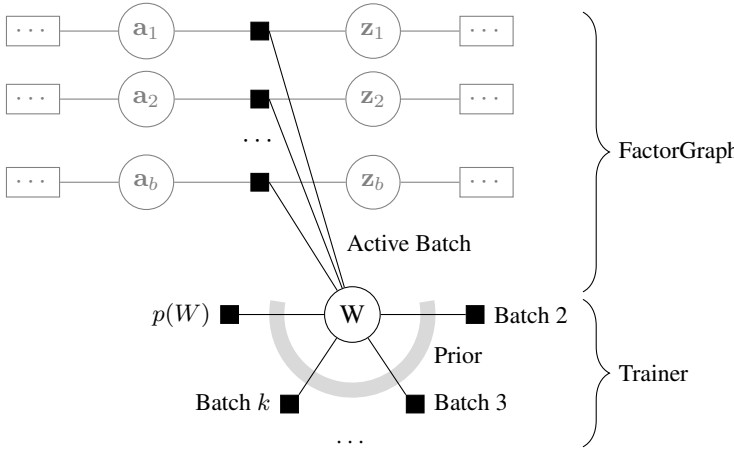

Figure 2: A full FactorGraph models all messages for one batch of training examples. To iterate the FactorGraph, we only need one joint message summarizing the prior and all other examples. When switching to a new batch, we aggregate messages from the previous batch and store them in the Trainer.

**Batching** As the forward and backward messages depend on each other, we must store them to compute message updates during message passing. Updating our messages in a sweeping "pass" over a branch and running backward passes immediately after the forward pass on the same branch, allows us to store many messages only temporarily, reducing memory requirements. This schedule also ensures efficient propagation of updated messages despite the presence of loops. However, some messages must still be retained permanently[4], leading to significant memory demand when storing them for all $n$ training examples. To address this, we adopt a batching strategy: Instead of maintaining $n$ training branches simultaneously, we update the factor graph using a batch (subset) of $b$ examples at a time. The factor graph then models $b$ messages to the weights $W$, while the messages to $W$ from the remaining (inactive) examples are aggregated into batch-wise products and stored in a trainer object. Figure 2 illustrates this setup. When switching batches, we divide the marginals by the batch's old aggregate message and multiply the updated messages into the marginal, ensuring that data is not double-counted. Within each batch, we iterate through the examples and perform a forward and backward pass on each in sequence. After all examples have been processed once, we call it an "iteration". Depending on the training stage, we either repeat this process within the same batch or move to the next batch. As training progresses, we gradually increase the number of iterations per batch to allow for finer updates as the overall posterior comes closer to convergence.

**Prediction:** Ultimately, our goal is to compute the marginal of the unobserved target $\mathbf{y}$ for some unseen input $\mathbf{x}$. Since the prediction branch in Figure 1 introduces additional loops, obtaining an accurate approximation would require iterating over the entire factor graph, including the training branches. In neural network terms, this translates to retraining the whole network for every test input. Instead, we pass messages only on the training branches in the batch-wise setup described above. At test time, messages from the training branches are propagated to the prediction branch, but not vice versa. Specifically, messages from the weights to the prediction branch are computed as the

---

[4]For example, the backward message of the linear layer is needed to compute the marginal of the inputs, which the forward message depends on.

product of the prior and the incoming messages from the training branches. This can be interpreted as approximating the posterior over weights, $p(\theta \mid \mathcal{D})$, with a diagonal Gaussian $\check{q}(\theta)$ and using it as the prior during inference.

## 4 MAKING IT SCALE

In scaling our approach to deep networks, we encountered several challenges related to computational performance, numerical stability, and weight initialization. The following subsections detail remedies to these problems.

### 4.1 FACTOR GRAPH IMPLEMENTATION

While batching effectively reduces memory requirements for large datasets, a direct implementation of a factor graph still scales poorly for deep networks. Explicitly modeling each scalar variable and factor as an instance is computationally expensive. To address this, we propose the following design optimizations: ① Rather than modeling individual elements of the factor graph, we represent entire layers of the network. Message passing between layers is orchestrated by an outer training loop. ② Each layer instance operates across all training branches within the active batch, removing the need to duplicate layers for each example. ③ Factors are stateless functions, not objects. Each layer is responsible for computing its forward and backward messages by calling the required functions. In this design, layer instances maintain their own state, but message passing and batching are managed in the outer loop. The stateless message equations are optimized for both performance and numerical stability. As a result, the number of layer instances scales linearly with network depth but remains constant regardless of layer size or batch size. This approach significantly reduces computational and memory overhead—our implementation is approximately 300x faster than a direct factor graph model in our tests. Additionally, we optimized our implementation for GPU execution by leveraging Julia's `CUDA.jl` and `Tullio.jl` libraries. Since much of the runtime is spent on linear algebra operations (within linear or convolutional layers), we built a reusable, GPU-compatible library for Gaussian multiplication. This design makes the implementation both scalable and extendable.

### 4.2 NUMERICAL STABILITY

Maintaining numerical stability in the message-passing process is critical, particularly as model size increases. Backward messages often exhibit near-infinite variances when individual weights have minimal impact on the likelihood. Therefore, we compute them directly in natural parameters, which also simplifies the equations. Special care is needed for LeakyReLU, as its messages can easily diverge. To mitigate this, we introduced guardrails: when normalization constants become too small, precision turns negative, or variance in forward messages increases, we revert to either $\mathbb{G}(0,0)$ or use moment matching on messages instead of marginals (see E for details). Another trick is to periodically recompute the weight marginals from scratch to maintain accuracy. By leveraging the properties of Gaussians, we save memory by recomputing variable-to-factor messages as needed[5]. However, incremental updates to marginals can accumulates errors, so we perform a full recomputation once per batch iteration. Lastly, we apply light message damping through an exponential moving average to stabilize the training, but, importantly, only on the aggregated batch messages, not on the individual messages of the active batch.

### 4.3 WEIGHT PRIORS

A zero-centered diagonal Gaussian prior with variance $\sigma_p^2$ is a natural choice for the prior over weights. However, as in traditional deep learning, setting all means to zero prevents messages from breaking symmetry. To resolve this, we sample prior means according to spectral parametrization (Yang et al., 2024), which facilitates feature learning independent of the network width. Another challenge in prior choice is managing exploding variances. In a naive setup with $\sigma_p^2 = 1$, forward

---

[5]Each layer stores factor-to-weight-variable messages and the marginal, which is an aggregate that is continuously updated as individual messages change. To compute a variable-to-factor message, divide the marginal by the factor-to-variable message.

message variances grow exponentially with the network depth. While we attempted to find a principled choice of $\sigma_p^2$, our current initialization scheme is based on experimental data (see D). For a layer with $d_1$ inputs and $d_2$ outputs, we set

$$\sigma_p^2 = \frac{1.5 - 0.8041 \cdot \min(1.0, d_2/d_1)}{0.8041 + 0.4496 \cdot d_1}.$$

Refer to D for our justification of this formula.

## 5 EXPERIMENTS

### 5.1 SYNTHETIC DATA

We first evaluate our model on a synthetic sine curve dataset of 200 data points. Figure 3 shows that an MLP with 4-5 linear layers fits the data well, whereas smaller models are not expressive enough to capture the data and deeper models are harder to fit. For depths beyond six layers, the performance degrades further, but the same is true for models of the same architecture trained in Torch. As expected, the posterior approximations in Figure 3 have small variance within the training range and high variance outside or when the fit is bad. In all plots, the mean prediction and standard deviation expand linearly outside the training range.

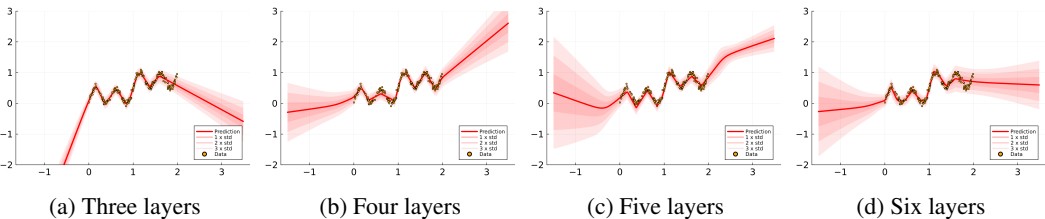

|        (a) Three layers        |        (b) Four layers        |        (c) Five layers        |        (d) Six layers        |

Figure 3: Fitting MLPs of width 16 with increasing depth. Between any linear layers we apply LeakyReLU with a leak of $0.1$. As the depth increases, the network becomes more expressive but harder to fit.

To assess how well our model's posterior uncertainty generalizes beyond the training data, we trained 100 separate models on the same sine curve data and evaluated their performance on unseen inputs. For this test, we limit the training data range to $(-0.5, 0.5)$ and then measure if the posterior approximation covers the true data-generating function outside of this training range. For negative $x$, 61% of $1\sigma$-intervals covered the true data-generating function, 86% of $2\sigma$-intervals, and 93% of $3\sigma$-intervals. For positive $x$, we measured 36%, 68%, and 90% respectively. While these measurements are slightly lower than the probability mass covered by the respective intervals, the posterior uncertainty appears to be reasonably well-calibrated. Overall, we found a strong correlation of 0.90 between credible intervals of the predictive posterior and the coverage rate.

### 5.2 CIFAR-10

To evaluate our method on the CIFAR-10 dataset we trained a 6 layer deep convolutional network with roughly 890k parameters on the full training dataset. As baseline methods we picked the SOTA optimizers AdamW (Loshchilov & Hutter, 2017) and IVON (Shen et al., 2024) each with a cosine annealing learning rate schedule (Loshchilov & Hutter, 2016). Across all methods, including ours, we trained for 25 epochs. In Appendix C we give extensive details on the network architecture and the experimental setup in general. Table 1 compares the performance of our method (MP) against AdamW and IVON across a variety of standard metrics. In general, we see that MP can compete with these two strong baselines. And in the expected calibration error our method even has a notable edge. That the metrics are overall worse than what is reported by Shen et al. (2024) is likely due to a difference in architecture; Shen et al. only conduct experiments on ResNets equipped with filter response normalization (Singh & Krishnan, 2019). Neither residual connections nor normalization layers are yet implemented in our factor graph library. Nevertheless, these results motivate to further improve our approach. In the future work part of Section 6 we outline ideas on how to model such factors.

|  | Acc. ↑ | Top-5 Acc. ↑ | NLL ↓ | ECE ↓ | Brier ↓ | OOD-AUROC ↑ |
|---|---|---|---|---|---|---|
| AdamW | **0.783** | **0.984** | 1.736 | 0.046 | 0.38 | 0.792 |
| IVON@mean | 0.772 | 0.983 | 1.494 | 0.041 | 0.387 | **0.819** |
| IVON | 0.772 | 0.983 | 1.316 | 0.035 | 0.37 | 0.808 |
| MP (Ours) | 0.773 | 0.977 | **0.997** | **0.029** | **0.361** | 0.81 |

Table 1: Comparison of various validation statistics for a convolutional network of roughly 890k parameters trained on CIFAR-10. Out-of-distribution (OOD) detection was tested with SVHN. For IVON we used 100 samples for prediction at test time. IVON@mean are the results obtained from evaluating the model at the means of the learned distributions of the individual parameters.

**Reproducibility** All code is available at https://github.com/iclr2025-7302/iclr2025_7302.

# 6 CONCLUSION

**Summary:** We presented a novel framework that advances message-passing (MP) for Bayesian neural networks by modeling the predictive posterior as a factor graph. To the best of our knowledge, this is the first MP method to handle convolutional neural networks while avoiding double-counting training data, a limitation in previous MP approaches like Soudry et al. (2014); Hernández-Lobato & Adams (2015); Lucibello et al. (2022). In our experiment on the CIFAR-10 dataset our method proofed to be competitive with the SOTA baselines AdamW and IVON, even showing an edge in terms of calibration.

**Limitations:** Despite recent advances, variational inference methods like IVON remain ahead in scale and performance on larger datasets. Our approach's runtime and memory requirements scale linearly with model parameters and dataset size. While our inference at test time can keep up with IVON's sampling approach in terms of speed and memory requirements, training is up to two orders of magnitude slower and more GPU-memory intensive compared to training deterministic networks using PyTorch with optimizers like AdamW.

The memory overhead stems from two key factors: First, each training example stores messages proportional to the model's parameter count, unlike AdamW's batch-level intermediate representations. Second, each parameter requires two 8-byte floating-point numbers, contrasting with more efficient 4-byte or smaller formats.

Runtime inflation results from several performance bottlenecks: Our training schedule lacks parallel forward passes, our Tullio-based CUDA kernel generation misses memory-layout and GPU optimizations present in mature libraries like Torch, message equations involve complex computations beyond standard matrix multiplications, and we use Julia's default FP64 precision, which GPUs process less efficiently.

**Future Work:** We believe Moment Propagation (MP) holds significant promise for more balanced uncertainty estimates, thanks to its moment-matching ability, compared to Variational Inference's tendency toward overconfident predictions. Further improvements in scalability and architectural flexibility could make MP a competitive alternative to VI.

Concretely, in terms of memory requirements, it is worth exploring whether iterating on individual examples instead of batches, and starting from scratch in each epoch, could leave our method ahead. While this might reintroduce the double counting problem, it would drastically reduce the GPU-memory footprint. Regarding training efficiency, an altered message-update schedule with actual batched computations would significantly reduce training time. Reimplementing our library in CUDA C++ with efficiency in mind could also drastically cut down computational overhead.

On the architectural front, we deem it likely that our approach can be extended to most modern deep learning architectures. Residual connections are straightforward to implement as they boil down to simple sum factors. For normalization layers at the scalar level, only a division factor is missing, which can be approximated by a "rotated" product factor. This would suffice to model ResNet-like architectures and more modern convolutional networks like ConvNeXt. For transformers, the last

ingredient needed is an efficient softargmax factor. Given the division factor, only an exp factor is missing to model softargmax at the scalar level.

Finally, future work might also explore applications to more applied tasks such as continual learning, sparse networks, and Bayesian reinforcement learning.

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

## A   PROOF OF GLOBAL MINIMIZATION OBJECTIVE

### A.1   MOMENT-MATCHED GAUSSIANS MINIMIZE CROSS-ENTROPY

Consider a scalar density $p$ and a Gaussian $q(\theta) = \mathcal{N}(\theta, \mu, \sigma)$. Then

$$\min H(p, q) = \min \left( \int p(\theta) \log \left( \frac{p(\theta)}{q(\theta)} \right) d\theta \right) = \min \left( \frac{1}{2\sigma^2} \int p(\theta)(\theta - \mu)^2 \, d\theta + \frac{\log(2\pi\sigma^2)}{2} \right).$$

It is well known that expectations minimize the expected mean squared error. In other words, the integral is minimized by setting $\mu$ to the expectation of $p$ and is then equal to the variance of $p$. The necessary condition of a local minimum then yields that $\sigma^2$ must be the variance of $p$.

### A.2   PROOF OF EQUATION (4) GLOBAL MINIMIZATION OBJECTIVE

Let $p$ be an arbitrary probability density on $\mathbb{R}^k$ with marginals $p_i(\theta_i) := \int p(\theta) \, d(\theta \setminus \theta_i)$ and denote by $\mathcal{Q}$ the set of diagonal Gaussians. Then for every $q(\boldsymbol{\theta}) = \prod_{i=1}^{k} q_i(\theta_i) \in \mathcal{Q}$ we can write the relative entropy from $p$ to $q$ as

$$D_{\mathrm{KL}}[\,p\,||\,q\,] = \int p(\boldsymbol{\theta}) \log \left( \frac{p(\boldsymbol{\theta})}{q(\boldsymbol{\theta})} \right) d\boldsymbol{\theta} = -\sum_{i=1}^{k} \int p(\boldsymbol{\theta}) \log(q(\theta_i)) d\boldsymbol{\theta} - H(p)$$

$$= -\sum_{i=1}^{k} \int_{\theta_i} \log(q_i(\theta_i)) \int_{\boldsymbol{\theta} \setminus \theta_i} p(\boldsymbol{\theta}) d(\boldsymbol{\theta} \setminus \theta_i) - H(p) = \sum_{i=1}^{k} H(p_i, q_i) - H(p).$$

This shows that $D_{\mathrm{KL}}[\,p\,||\,q\,]$ is minimized by independently minimizing the summands $H(p_i, q_i)$. In combination with A.1 this completes the proof.

## B   DERIVATIONS OF MESSAGE EQUATIONS

### B.1   RELU

A common activation function is the Rectified Linear Unit $\mathrm{ReLU} : \mathbb{R} \to \mathbb{R}, z \mapsto \max(0, z)$.

**Forward Message:**   Since ReLU is not injective, we cannot apply the density transformation property of the Dirac delta to the forward message

$$m_{f \to a}(\mathrm{a}) = \int_{\mathrm{z} \in \mathbb{R}} \delta(\mathrm{a} - \mathrm{ReLU(z)}) m_{\mathrm{z} \to f}(\mathrm{z}) \, d\mathrm{z}.$$

In fact, the random variable $\mathrm{ReLU}(Z)$ with $Z \sim m_{z \to f}$ does not even have a density. A positive amount of weight, namely $\Pr[Z \leq 0]$, is mapped to 0. Therefore

$$m_{f \to a}(0) = \lim_{t \to 0} \int_{\mathrm{z} \in \mathbb{R}} \mathcal{N}(\mathrm{ReLU(z)}; 0, t^2) m_{\mathrm{z} \to f}(\mathrm{z}) \, d\mathrm{z} \geq \lim_{t \to 0} \mathcal{N}(0; 0, t^2) \min_{\mathrm{z} \in [-1, 0]} m_{\mathrm{z} \to f}(\mathrm{z}) = \infty.$$

Apart from 0, the forward message is well defined everywhere, and technically null sets do not matter under the integral. However, moment-matching $m_{z \to f}$ while truncating at 0 does not seem reasonable as it completely ignores the weight of $m_{z \to f}$ on $\mathbb{R}_{\leq 0}$. Therefore, we refrain from moment-matching the forward message of ReLU.

As an alternative, we consider a marginal approximation. That means, we derive formulas for

$$m_k := \int_{a \in \mathbb{R}} a^k m_{a \to f}(a) m_{f \to a}(a) \, da, \quad k \in \{0, 1, 2\} \tag{5}$$

and then set

$$m_{f \to a}(a) := \mathcal{N}(a; m_1/m_0, m_2/m_0 - (m_1/m_0)^2) / m_{a \to f}(a).$$

By changing the integration order, we obtain

$$m_k = \int_{a \in \mathbb{R}} a^k m_{a \to f}(a) \int_{z \in \mathbb{R}} \delta(a - \mathrm{ReLU}(z)) m_{z \to f}(z) \, dz \, da$$

$$= \int_{z \in \mathbb{R}} m_{z \to f}(z) \int_{a \in \mathbb{R}} \delta(a - \mathrm{ReLU}(z)) a^k m_{a \to f}(a) \, da \, dz$$

$$= \int_{z \in \mathbb{R}} m_{z \to f}(z) \mathrm{ReLU}^k(z) m_{a \to f}(\mathrm{ReLU}(z)) \, dz$$

Note that we end up with a well-defined and finite integral. Similar integrals arise in later derivations. For this reason we encapsulate part of the analysis in basic building blocks.

*Building Block* 1. We can efficiently approximate integrals of the form

$$\int_0^\infty z^k \mathcal{N}(z; \mu_1, \sigma_1^2) \mathcal{N}(z; \mu_2, \sigma_2^2) \, dz$$

where $\mu_1, \mu_2 \in \mathbb{R}, \sigma_1, \sigma_2 > 0$ and $k = 0, 1, 2$.

*Proof.* By Equation (3) the integral is equal to

$$S^+ = \mathcal{N}(\mu_1; \mu_2, \sigma_1^2 + \sigma_2^2) \int_0^\infty z^k \mathcal{N}\left(z; \mu, \sigma^2\right) \, dz$$

$$= \mathcal{N}(\mu_1; \mu_2, \sigma_1^2 + \sigma_2^2) \begin{cases} \mathbb{E}[\mathrm{ReLU}^k(\mathcal{N}(\mu, \sigma^2))] & \text{for } k = 1, 2 \\ \Pr[-Z \le 0] = \phi(\mu/\sigma) & \text{for } k = 0 \end{cases}$$

where

$$\mu = \frac{\tau}{\rho}, \quad \sigma^2 = \frac{1}{\rho}, \quad \tau = \frac{\mu_1}{\sigma_1^2} + \frac{\mu_2}{\sigma_2^2} \quad \text{and} \quad \rho = \frac{1}{\sigma_1^2} + \frac{1}{\sigma_2^2}.$$

$\square$

This motivates the derivation of efficient formulas for the moments of an image of a Gaussian variable under ReLU.

*Building Block* 2. Let $Z \sim \mathcal{N}(\mu, \sigma^2)$. The first two moments of ReLU(Z) are then given by

$$\mathbb{E}[\mathrm{ReLU}(Z)] = \sigma \varphi(x) + \mu \phi(x) \tag{6}$$

$$\mathbb{E}[\mathrm{ReLU}^2(Z)] = \sigma \mu \varphi(x) + (\sigma^2 + \mu^2) \phi(x), \tag{7}$$

where $x = \mu/\sigma$ and $\varphi, \phi$ denote the pdf and cdf of the standard normal distribution, respectively.

*Proof.* The basic idea is to apply $\int z e^{-z^2/2} \, dz = -e^{-z^2/2}$. Together with a productive zero, one obtains

$$\sqrt{2\pi} \sigma \mathbb{E}[\mathrm{ReLU}(Z)] = \int_0^\infty z e^{-\frac{(z-\mu)^2}{2\sigma^2}} \, dz = \sigma^2 \int_0^\infty \frac{(z-\mu)}{\sigma^2} e^{-\frac{(z-\mu)^2}{2\sigma^2}} \, dz + \mu \int_0^\infty e^{-\frac{(z-\mu)^2}{2\sigma^2}} \, dz$$

$$= \sigma^2 \left[ -e^{-\frac{(z-\mu)^2}{2\sigma^2}} \right]_0^\infty + \sqrt{2\pi} \sigma \mu \Pr[Z \ge 0]$$

$$= \sigma^2 e^{-\frac{\mu^2}{2\sigma^2}} + \sqrt{2\pi} \sigma \mu \Pr\left[ \frac{-Z + \mu}{\sigma} \le \frac{\mu}{\sigma} \right]$$

$$= \sqrt{2\pi} \sigma^2 \varphi(x) + \sqrt{2\pi} \sigma \mu \phi(x).$$

Rearranging yields the desired formula for the first moment. For the second moment, we need to complete the square and perform integration by parts:

$$\mathbb{E}[\text{ReLU}^2(Z)] = \frac{1}{\sqrt{2\pi}\sigma} \int_0^\infty z^2 e^{-\frac{(z-\mu)^2}{2\sigma^2}} \, dz$$

$$= \frac{1}{\sqrt{2\pi}\sigma} \left( \sigma^2 \int_0^\infty (z-\mu)\frac{z-\mu}{\sigma^2} e^{-\frac{(z-\mu)^2}{2\sigma^2}} \, dz + 2\mu \int_0^\infty z e^{-\frac{(z-\mu)^2}{2\sigma^2}} \, dz - \mu^2 \int_0^\infty e^{-\frac{(z-\mu)^2}{2\sigma^2}} \, dz \right)$$

$$= \frac{\sigma^2}{\sqrt{2\pi}\sigma} \left( \left[ -(z-\mu)e^{-\frac{(z-\mu)^2}{2\sigma^2}} \right]_0^\infty + \int_0^\infty e^{-\frac{(z-\mu)^2}{2\sigma^2}} \right) + 2\mu\mathbb{E}[\text{ReLU}(Z)] - \mu^2\phi(x)$$

$$= -\sigma\mu\varphi(x) + \sigma^2\phi(x) + 2\mu\mathbb{E}[\text{ReLU}(Z)] - \mu^2\phi(x) = \sigma\mu\varphi(x) + (\sigma^2 + \mu^2)\phi(x).$$

$\square$

*Building Block* 3. Integrals of the form

$$S^- := \int_{-\infty}^0 z^k \mathcal{N}(z; \mu_1, \sigma_1^2)\mathcal{N}(0; \mu_2, \sigma_2^2) \, dz$$

where $\mu_1, \mu_2 \in \mathbb{R}, \sigma_1, \sigma_2 > 0$ and $k = 0, 1, 2$ can be efficiently approximated.

*Proof.* Employing the substitution $z = -t$ gives

$$S^- = \mathcal{N}(0; \mu_2, \sigma_2^2) \int_0^\infty (-1)^k t^k \mathcal{N}(-t; \mu_1, \sigma_1^2) \, dt = (-1)^k \mathcal{N}(0; \mu_2, \sigma_2^2) \int_0^\infty t^k \mathcal{N}(t; -\mu_1, \sigma_1^2) \, dt$$

$$= (-1)^k \mathcal{N}(0; \mu_2, \sigma_2^2) \begin{cases} \mathbb{E}[\text{ReLU}(\mathcal{N}(-\mu_1, \sigma_1^2))] & \text{for } k = 1, 2 \\ \Pr[-Z \geq 0] = \phi(-\mu_1/\sigma_1) & \text{for } k = 0. \end{cases}$$

$\square$

Now let $m_{z \to f}(z) = \mathcal{N}(z; \mu_z, \sigma_z^2), m_{a \to f}(a) = \mathcal{N}(a; \mu_a, \sigma_a^2)$ and consider the decomposition

$$m_k = \underbrace{\int_0^\infty z^k \mathcal{N}(z; \mu_z, \sigma_z^2)\mathcal{N}(z; \mu_a, \sigma_a^2) \, dz}_{S^+} + \underbrace{\int_{-\infty}^0 \text{ReLU}^k(z)\mathcal{N}(z; \mu_z, \sigma_z^2)\mathcal{N}(0; \mu_a, \sigma_a^2) \, dz}_{S^-}.$$

Note that $S^+$ falls under Building Block 1 for any $k = 0, 1, 2$. The other addend $S^-$ is equal to 0 for $k = 1, 2$, and is handled by Building Block 3 for $k = 0$.

**Backward Message:** By definition of the Dirac delta, the backward message is equal to

$$m_{f \to z}(z) = \int_{a \in \mathbb{R}} \delta(a - \text{ReLU}(z)) m_{a \to f}(a) \, da = m_{a \to f}(\text{ReLU}(z))$$

which is, of course, not integrable, so it cannot be interpreted as a scaled density. Instead, we apply marginal approximation by deriving formulas for

$$m_k := \int_{z \in \mathbb{R}} z^k m_{z \to f}(z) m_{f \to z}(z) \, dz, \quad k \in \{0, 1, 2\}$$

and then setting

$$m_{f \to z}(z) := \mathcal{N}(z; m_1/m_0, m_2/m_0 - (m_1/m_0)^2) \, / \, m_{z \to f}(z).$$

To this end, let $m_{z \to f}(z) = \mathcal{N}(z; \mu_z, \sigma_z^2)$ and $m_{a \to f}(a) = \mathcal{N}(a; \mu_a, \sigma_a^2)$. Then we have

$$m_k = \underbrace{\int_0^\infty z^k \mathcal{N}(z; \mu_z, \sigma_z^2)\mathcal{N}(z; \mu_a, \sigma_a^2) \, dz}_{S^+} + \underbrace{\int_{-\infty}^0 z^k \mathcal{N}(z; \mu_z, \sigma_z^2)\mathcal{N}(0; \mu_a, \sigma_a^2) \, dz}_{S^-}.$$

The two addends $S^+$ and $S^-$ are handled by Building Block 1 and Building Block 3, respectively.

## B.2 LEAKY ReLU

Another common activation function is the Leaky Rectified Linear Unit

$$\text{LeakyReLU}_\alpha : \mathbb{R} \to \mathbb{R}, z \mapsto \begin{cases} z & \text{for } z \geq 0 \\ \alpha z & \text{for } z < 0. \end{cases}$$

It is parameterized by some $\alpha > 0$ that is typically small, such as $\alpha = 0.1$. In contrast to ReLU, it is injective (and even bijective). For this reason the forward and backward messages are both integrable and can be approximated by both direct and marginal moment matching. The notation is shown in Figure 4.

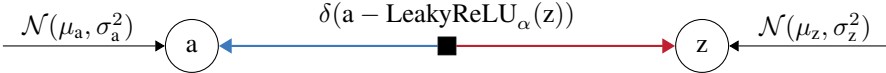

Figure 4: A deterministic factor corresponding to the $\text{LeakyReLU}_\alpha$ activation function.

**Forward Message:** It is easy to show that the density of $\text{LeakyReLU}_\alpha(\mathcal{N}(\mu_z, \sigma_z^2))$ is given by

$$p(a) = \mathcal{N}(\text{LeakyReLU}_{1/\alpha}(a); \mu_z, \sigma_z^2) \begin{cases} 1 & \text{for } z \geq 0 \\ 1/\alpha & \text{for } z < 0 \end{cases}$$

which only has one discontinuity point, namely 0. In particular, it is continuous almost everywhere. So by the density transformation property of Dirac's delta, we have $m_{f \to a}(a) = p(a)$ for almost all $a$. Under the integral we can therefore replace $m_{f \to a}(a)$ by $p(a)$. This justifies that the moments of $m_{f \to a}$ are exactly the moments of $(\text{LeakyReLU}_\alpha)_* \mathcal{N}(\mu_z, \sigma_z^2)$. Its expectation is equal to

$$\mathbb{E}\left[\text{LeakyReLU}_\alpha(\mathcal{N}(\mu_z, \sigma_z^2))\right] = \int_{-\infty}^0 \alpha z \mathcal{N}(z; \mu_z, \sigma_z^2)\, dz + \int_0^\infty z \mathcal{N}(z; \mu_z, \sigma_z^2)\, dz$$

$$= -\alpha \int_0^\infty t \mathcal{N}(t; -\mu_z, \sigma_z^2)\, dt + \int_0^\infty z \mathcal{N}(z; \mu_z, \sigma_z^2)\, dz$$

$$= -\alpha \mathbb{E}[\text{ReLU}(\mathcal{N}(-\mu_z, \sigma_z^2))] + \mathbb{E}[\text{ReLU}(Z)].$$

Both addends are handled by Building Block 2. Yet we can get more insight by further substitution:

$$\mathbb{E}[\text{LeakyReLU}_\alpha(Z)] = -\alpha(\sigma_z \varphi(-\mu_z/\sigma_z) - \mu_z \phi(-\mu_z/\sigma_z)) + \sigma_z \varphi(\mu_z/\sigma_z) + \mu_z \phi(\mu_z/\sigma_z)$$

$$= (1-\alpha)(\sigma_z \varphi(\mu_z/\sigma_z) + \mu_z \phi(\mu_z/\sigma_z)) + \alpha \mu_z$$

$$= (1-\alpha)\mathbb{E}[\text{ReLU}(Z)] + \alpha \mathbb{E}[Z].$$

In the second to last equation, we use the identities $\varphi(-x) = \varphi(x)$ and $\phi(-x) = 1 - \phi(x)$. As such, the mean of $\text{LeakyReLU}_\alpha(Z)$ is a convex combination of the mean of $\text{ReLU}(Z)$ and the mean of $Z$. The function $\text{LeakyReLU}_1$ the identity, and its mean is accordingly the mean of $Z$. For $\alpha = 0$, we recover the mean of $\text{ReLU}(Z)$.

The second moment of $\text{LeakyReLU}_\alpha(Z)$ decomposes to

$$\mathbb{E}[\text{LeakyReLU}_\alpha^2(Z)] = \int_{-\infty}^0 \alpha^2 z^2 \mathcal{N}(z; \mu_z, \sigma_z^2)\, dz + \int_0^\infty z^2 \mathcal{N}(z; \mu_z, \sigma_z^2)\, dz$$

$$= \alpha^2 \int_0^\infty z^2 \mathcal{N}(z; -\mu_z, \sigma_z^2)\, dz + \int_0^\infty z^2 \mathcal{N}(z; \mu_z, \sigma_z^2)\, dz$$

$$= \alpha^2 \mathbb{E}[\text{ReLU}^2(\mathcal{N}(-\mu_z, \sigma_z^2))] + \mathbb{E}[\text{ReLU}^2(\mathcal{N}(\mu_z, \sigma_z^2))].$$

Again, both addends are covered by Building Block 2, so approximating the forward message via direct moment matching is feasible.

A marginal approximation can also be found. For all $k = 0, 1, 2$ we have

$$\int_{a \in \mathbb{R}} a^k m_{a \to f}(a) m_{f \to a}(a) \, da = \int_{a \in \mathbb{R}} a^k m_{a \to f}(a) p(a) \, da$$

$$= \frac{1}{\alpha} \underbrace{\int_{-\infty}^0 a^k \mathcal{N}(a; \mu_a, \sigma_a^2) \mathcal{N}(a/\alpha; \mu_z, \sigma_z^2) \, da}_{S^-} + \underbrace{\int_0^\infty a^k \mathcal{N}(a; \mu_a, \sigma_a^2) \mathcal{N}(a; \mu_z, \sigma_z^2) \, da}_{S^+}$$

The term $S^+$ is handled by Building Block 1. The term $S^-$ is equal to

$$S^- = \int_{-\infty}^0 a^k \mathcal{N}(a; \mu_a, \sigma_a^2) \mathcal{N}(a; \alpha\mu_z, (\alpha\sigma_z)^2) \, da$$

$$= (-1)^k \int_0^\infty a^k \mathcal{N}(a; -\mu_a, \sigma_a^2) \mathcal{N}(a; -\alpha\mu_z, (\alpha\sigma_z)^2) \, da$$

and therefore also covered by Building Block 1.

**Backward Message:**  By the sifting property of the Dirac delta, the backward message is equal to

$$m_{f \to z}(z) = \int_{a \in \mathbb{R}} \delta(a - \text{LeakyReLU}_\alpha(z)) m_{a \to f}(a) \, da = m_{a \to f}(\text{LeakyReLU}_\alpha(z)).$$

As opposed to ReLU, the backward message is integrable. That means, we can also apply direct moment matching: For all $k = 0, 1, 2$ we have

$$m_{f \to z}(z) = \int_{-\infty}^0 z^k \mathcal{N}(\alpha z; \mu_a, \sigma_a^2) \, dz + \int_0^\infty z^k \mathcal{N}(z; \mu_a, \sigma_a^2) \, dz$$

$$= \frac{(-1)^k}{\alpha} \int_0^\infty z^k \mathcal{N}(z; -\mu_a/\alpha, (\sigma_a/\alpha)^2) \, dz + \int_0^\infty z^k \mathcal{N}(z; \mu_a, \sigma_a^2) \, dz$$

For $k = 1$ or $k = 2$, the integrals fall under Building Block 2 again. If $k = 0$, then

$$m_{f \to z}(z) = \frac{(-1)^k}{\alpha} \phi(-\mu_a/\sigma_a) + \phi(\mu_a/\sigma_a).$$

Again, we can also find a marginal approximation as well. For all $k = 0, 1, 2$, we can write

$$\int_{z \in \mathbb{R}} z^k m_{z \to f}(z) m_{f \to z}(z) \, dz$$

$$= \int_{-\infty}^0 z^k \mathcal{N}(z; \mu_z, \sigma_z^2) \mathcal{N}(\alpha z; \mu_a, \sigma_a^2) \, dz + \int_0^\infty z^k \mathcal{N}(z; \mu_z, \sigma_z^2) \mathcal{N}(z; \mu_a, \sigma_a^2) \, dz$$

$$= \frac{(-1)^k}{\alpha} \int_0^\infty z^k \mathcal{N}(z; -\mu_z, \sigma_z^2) \mathcal{N}(z; -\mu_a/\alpha, (\sigma_a/\alpha)^2) \, dz + \int_0^\infty z^k \mathcal{N}(z; \mu_z, \sigma_z^2) \mathcal{N}(z; \mu_a, \sigma_a^2) \, dz$$

Since both integrals are covered by Building Block 1 we have derived direct and marginal approximations of LeakyReLU messages using moment matching.

B.3 SOFTMAX

We model the soft(arg)max training signal as depicted in Table 3. For the forward message on the prediction branch, we employ the so-called "probit approximation" (Daxberger et al., 2022):

$$m_{f \to c}(i) = \int \text{softmax}(\mathbf{a})_i \mathcal{N}(\mathbf{a}; \boldsymbol{\mu}, \text{diag}(\boldsymbol{\sigma}^2) \, d\mathbf{a} \approx \text{softmax}(\boldsymbol{t})_i,$$

where $t_j = \mu_j / (1 + \frac{\pi}{8} \sigma_j^2)$, $j = 1, \ldots, d$. For the backward message on a training branch, to say $a_d$, we use marginal approximation. We hence need to compute the moments $m_0, m_1, m_2$ of the marginal of $a_d$ via:

$$m_k = \int a_d^k \, \text{softmax}(\mathbf{a})_c \, \mathcal{N}(\mathbf{a}; \boldsymbol{\mu}, \text{diag}(\boldsymbol{\sigma}^2) \, d\mathbf{a}$$

$$= \int_{a_d} a_d^k \mathcal{N}(a_d; \mu_d, \sigma_d^2) \int_{\mathbf{a} \backslash a_d} \text{softmax}(\mathbf{a})_i \prod_{j \neq i} \mathcal{N}(a_j; \mu_j, \sigma_j^2) \, d(\mathbf{a} \backslash a_d) da_d.$$

We can reduce the inner integral to the probit approximation by regarding the point distribution $\delta_{a_d}$ as the limit of a Gaussian with vanishing variance:

$$\int_{\mathbf{a}\backslash a_d} \text{softmax}(\mathbf{a})_c \prod_{j\neq d} \mathcal{N}(a_j; \mu_j, \sigma_j^2) \, d(\mathbf{a}\backslash a_d)$$

$$= \int_{\mathbf{a}\backslash a_d} \int_{\tilde{a}_d} \delta(\tilde{a}_d - a_d) \, \text{softmax}(a_1, \ldots, a_{d-1}, \tilde{a}_d) \prod_{j\neq d} \mathcal{N}(a_j; \mu_j, \sigma_j^2) \, d\tilde{a}_d \, d(\mathbf{a}\backslash a_d)$$

$$= \int_{\tilde{\mathbf{a}}\backslash \tilde{a}_d} \lim_{\sigma\to 0} \int_{\tilde{a}_i} \text{softmax}(\tilde{\mathbf{a}})_c \, \mathcal{N}(\tilde{a}_d; a_d, \sigma^2) \prod_{j\neq d} \mathcal{N}(\tilde{a}_j; \mu_j, \sigma_j^2) \, d\tilde{a}_i \, d\tilde{\mathbf{a}}_i$$

By Lebesgue's dominated convergence theorem we obtain equality to

$$\lim_{\sigma\to 0} \int_{\tilde{\mathbf{a}}} \text{softmax}(\tilde{\mathbf{a}})_c \mathcal{N}(\tilde{a}_d; a_d, \sigma^2) \prod_{j\neq i} \mathcal{N}(\tilde{a}_j; \mu_j, \sigma_j^2) \, d\tilde{\mathbf{a}}$$

$$\approx \lim_{\sigma\to 0} \text{softmax}(\boldsymbol{t})_i = \text{softmax}(t_1, \ldots, t_{d-1}, a_d) \quad \text{where} \quad t_j = \begin{cases} \mu_j/(1 + \frac{\pi}{8}\sigma_j^2) & \text{for } j\neq d \\ a_d/(1 + \frac{\pi}{8}\sigma^2) & \text{for } j = d. \end{cases}$$

Hence, we can approximate $m_k$ by one-dimensional numerical integration of

$$m_k \approx \int_{a_d} a_d^k \, \mathcal{N}(a_d; \mu_d, \sigma_d^2) \, \text{softmax}(t_1, \ldots, t_{d-1}, a_d) \, da_d.$$

## C    EXPERIMENTAL SETUP

**Synthetic Data - Depth Scaling:**    We generated a dataset of 200 points by randomly sampling $x$ values from the range $[0, 2]$. The true data-generating function was

$$f(x) = 0.5x + 0.2\sin(2\pi \cdot x) + 0.3\sin(4\pi \cdot x).$$

The corresponding $y$ values were sampled by adding Gaussian noise: $f(x) + \mathcal{N}(0, 0.05^2)$. For the architecture, we used a three-layer neural network with the structure:

$$[\text{Linear}(1, 16), \text{LeakyReLU}(0.1), \text{Linear}(16, 16), \text{LeakyReLU}(0.1), \text{Linear}(16, 1)].$$

A four-layer network has one additional $[\text{Linear}(16, 16), \text{LeakyReLU}(0.1)]$ block in the middle, and a five-layer network has two additional blocks. For the regression noise hyperparameter, we used the true noise $\beta^2 = 0.05^2$. The models were trained for 500 iterations over one batch (as all data was processed in a single active batch).

**Synthetic Data - Uncertainty Evaluation:**    The same data-generation process was used as in the depth-scaling experiment, but this time, $x$ values were drawn from the range $[-0.5, 0.5]$. The network architecture remained the same as the three-layer network, but the width of the layers was increased to 32. We trained 100 networks with different random seeds on the same dataset. We define a $p$-credible interval for $0 \leq p \leq 1$ as:

$$[\text{cdf}^{-1}(0.5 - \frac{p}{2}), \text{cdf}^{-1}(0.5 + \frac{p}{2})].$$

For each credible interval mass $p$ (ranging from 0 to 1 in steps of 0.01), we measured how many of the $p$-credible intervals (across the 100 posterior approximations) covered the true data-generating function. This evaluation was done at each possible $x$ value (ranging from -20 to 20 in steps of 0.05), generating a coverage rate for each combination of $p$ and $x$. For each $p$, we then computed the median for $x > 10$ and the median for $x < -10$. If we correlate the $p$ values with the medians, we found that for the median obtained from positive $x$ values the correlation was 0.96, for negative $x$ it was 0.99, and for the combined set of medians it was 0.9.

**CIFAR-10:**    For our CIFAR-10 experiments, we used the default train-test split and trained the following feed-forward network:

```
class Net(nn.Module):
    def __init__(self):
        super(Net, self).__init__()
        self.model = nn.Sequential(
            # Block 1
            nn.Conv2d(3, 32, 3, padding=0),
            nn.LeakyReLU(0.1),
            nn.Conv2d(32, 32, 3, padding=0),
            nn.LeakyReLU(0.1),
            nn.MaxPool2d(2),
            # Block 2
            nn.Conv2d(32, 64, 3, padding=0),
            nn.LeakyReLU(0.1),
            nn.Conv2d(64, 64, 3, padding=0),
            nn.LeakyReLU(0.1),
            nn.MaxPool2d(2),
            # Head
            nn.Flatten(),
            nn.Linear(64 * 5 * 5, 512),
            nn.LeakyReLU(0.1),
            nn.Linear(512, 10),
        )

    def forward(self, x):
        return self.model(x)
```

In the case of AdamW and IVON we trained with a cross-entropy loss on the softargmax of the network output. For our message passing method we used our argmax factor as a training signal instead of softargmax, see Appendix E. The reason is that for softargmax we only have message approximations relying on rather expensive numerical integration. In our library this factor graph can be constructed via

```
fg = create_factor_graph([
            size(d.X_train)[1:end-1], # (3, 32, 32)
            # First Block
            (:Conv, 32, 3, 0), # (32, 30, 30)
            (:LeakyReLU, 0.1),
            (:Conv, 32, 3, 0), # (32, 28, 28)
            (:LeakyReLU, 0.1),
            (:MaxPool, 2), # (32, 14, 14)
            # Second Block
            (:Conv, 64, 3, 0), # (64, 12, 12)
            (:LeakyReLU, 0.1),
            (:Conv, 64, 3, 0), # (64, 10, 10)
            (:LeakyReLU, 0.1),
            (:MaxPool, 2), # (64, 5, 5)
            # Head
            (:Flatten,), # (64*5*5 = 1600)
            (:Linear, 512), # (512)
            (:LeakyReLU, 0.1),
            (:Linear, 10), # (10)
            (:Argmax, true)
        ], batch_size)
```

For all methods we used a batch size of 128 and trained for 25 epochs with a cosine annealing learning rate schedule. Concerning hyperparameters: For AdamW we found the standard parameters of $\mathrm{lr} = 10^{-3}, \beta_1 = 0.9, \beta_2 = 0.999, \epsilon = 10^{-8}$ and $\delta = 10^{-4}$ to work best. For IVON we followed the practical guidelines given in the Appendix of Shen et al. (2024).

To measure calibration, we used 20 bins that were split to minimize within-bin variance. For OOD recognition, we predicted the class of the test examples in CIFAR-10 (in-distribution) and SVHN (OOD) and computed the entropy over softmax probabilities for each example. We then sort them by negative entropy and test the true positive and false positive rates for each possible (binary) decision threshold. The area under this ROC curve is computed in the same way as for relative calibration.

# D   PRIOR ANALYSIS

The strength of the prior determines the amount of data needed to obtain a useful posterior that fits the data. Our goal is to draw prior means and set prior variances so that the computed variances of all messages are on the order of $\mathcal{O}(1)$ regardless of network width and depth. It is not entirely clear if this would be a desirable property; after all, adding more layers also makes the network more expressive and more easily able to model functions with very high or low values. However, if we let the predictive prior grow unrestricted, it will grow exponentially, leading to numerical issues. In the following, we analyze the predictive prior under simplifying assumptions to derive a prior initialization that avoids exponential variance explosion. While we fail to achieve this goal, our current prior variances are still informed by this analysis.

In the following, we assume that the network inputs are random variables. Then, the parameters of messages also become random variables, as they are derived from the inputs according to the message equations. Our goal is to keep the expected value of the variance parameter of the outgoing message at a constant size. We also assume that the means of the prior are sampled according to spectral initialization, as described in Section 4.3.

FIRSTGAUSSIANLINEARLAYER - INPUT IS A CONSTANT

Each linear layer transforms some $d_1$-dimensional input $\mathbf{x}$ to some $d_2$-dimensional output $\mathbf{y}$ according to $\mathbf{y} = W\mathbf{x} + \mathbf{b}$. In the first layer, $\mathbf{x}$ is the input data. For this analysis, we assume each element $x_i$ to be drawn independently from $x_i \sim \mathcal{N}(0, 1)$. Let $\mathbf{x}$ be a $d_1$-dimensional input vector, $\mathbf{m}_w$ be the prior messages from one column of $W$, and $z = \mathbf{w}'\mathbf{x}$ be the vector product before adding the bias.

During initialization of the weight prior, we draw the prior means using spectral parametrization and set the prior variances to a constant:

$$m_{w_i} = \mathcal{N}(\mu_{w_i}, \sigma_w^2) \text{ with } \mu_{w_i} \sim \mathcal{N}(0, l^2),$$

$$l = \frac{1}{\sqrt{k}} \cdot \min(1, \sqrt{\frac{d_2}{d_1}}).$$

By applying the message equations, we then approximate the forward message to the output with a normal distribution

$$m_z = \mathcal{N}(\mu_z, \sigma_z^2).$$

Because $\sigma_z^2$ depends on the random variables $x_i$, it is also a random variable that follows a scaled chi-squared distribution

$$\sigma_z^2 = \sum_{i=1}^{d_1} x_i^2 \cdot \sigma_w^2$$

$$\sigma_z^2 \sim \chi_{d_1}^2 \cdot \sigma_w^2$$

and its expected value is

$$\mathbb{E}[\sigma_z^2] = d_1 \cdot \sigma_w^2.$$

We conclude that we can control the magnitude of the variance parameter by choosing $\mathbb{E}[\sigma_z^2]$ and setting $\sigma_w^2 = \frac{\mathbb{E}[\sigma_z^2]}{d_1}$.

GAUSSIANLINEARLAYER - INPUT IS A VARIABLE

In subsequent linear layers, the input $\mathbf{x}$ is not observed and we receive an approximate forward message that consists of independent normal distributions

$$m_{x_i} = \mathcal{N}(\mu_{x_i}, \sigma^2_{x_i}).$$

Following the message equations, the outgoing forward message to $z$ then has a variance

$$\sigma^2_z = \sum_{i=1}^{d_1} (\sigma^2_{x_i} + \mu^2_{x_i}) \cdot (\sigma^2_w + \mu^2_{w_i}) - (\mu^2_{x_i} * \mu^2_{w_i})$$

$$= \sum_{i=1}^{d_1} \underbrace{\sigma^2_{x_i} \cdot \sigma^2_w}_{\text{I}} + \underbrace{\sigma^2_{x_i} \cdot \mu^2_{w_i}}_{\text{II}} + \underbrace{\mu^2_{x_i} \cdot \sigma^2_w}_{\text{III}}$$

The layer's prior variance $\sigma^2_w$ is a constant, whereas all other elements are random variables according to our assumptions. To make further analysis tractable, we also have to assume that the variances $\sigma^2_{x_i}$ of the incoming forward messages are identical constants for all $i$, not random variables. We furthermore assume that the means are drawn i.i.d. from:

$$\mu_{w_i} \sim \mathcal{N}(0, l^2)$$
$$\mu_{x_i} \sim \mathcal{N}(\mu_{\mu_x}, \sigma^2_{\mu_x}).$$

The random variable $\sigma^2_z$ then follows a generalized chi-squared distribution

$$\sigma^2_z \sim \left( \sum_{i=1}^{d_1} \underbrace{\sigma^2_x \cdot l^2 \cdot \chi^2(1, 0^2)}_{\text{II}} + \underbrace{\sigma^2_w \cdot \sigma^2_{\mu_x} \cdot \chi^2(1, \mu^2_{\mu_x})}_{\text{III}} \right) + \underbrace{d_1 \cdot \sigma^2_w \cdot \sigma^2_x}_{\text{I}}$$

and its expected value is

$$\mathbb{E}[\sigma^2_z] = \left( \sum_{i=1}^{d_1} \sigma^2_x \cdot l^2 \cdot (1 + 0^2) + \sigma^2_w \cdot \sigma^2_{\mu_x} \cdot (1 + \mu^2_{\mu_x}) \right) + d_1 \cdot \sigma^2_w \cdot \sigma^2_x$$

$$= d_1 \cdot \left( \sigma^2_x \cdot l^2 + \sigma^2_w \cdot \sigma^2_{\mu_x} \cdot (1 + \mu^2_{\mu_x}) + \sigma^2_w \cdot \sigma^2_x \right)$$

$$= \underbrace{d_1 \cdot \sigma^2_x \cdot l^2}_{\text{II}} + \underbrace{d_1 \cdot (\sigma^2_{\mu_x} \cdot (1 + \mu^2_{\mu_x}) + \sigma^2_x) \cdot \sigma^2_w}_{\text{I+III}}.$$

As $\sigma^2_w$ has to be positive, we conclude that if we choose $\mathbb{E}[\sigma^2_z] > d_1 \cdot \sigma^2_x \cdot l^2$, then we can set

$$\sigma^2_w = \frac{\mathbb{E}[\sigma^2_z] - d_1 \cdot \sigma^2_x \cdot l^2}{d_1 \cdot (\sigma^2_{\mu_x} \cdot (1 + \mu^2_{\mu_x}) + \sigma^2_x)}.$$

We know (or choose) $d_1$, $l^2$, and $\mathbb{E}[\sigma^2_z]$, but we require values for $\sigma^2_x$, $\mu^2_{\mu_x}$, and $\sigma^2_{\mu_x}$ to be able to choose $\sigma^2_w$. We will find empirical values for these parameters in the next section.

EMPIRICAL PARAMETERS + LEAKYRELU

To inform the choice of the prior variances of the inner linear layers, we also need to analyze LeakyReLU. We assume the network is an MLP that alternates between linear layers and LeakyReLU. As the message equations of LeakyReLU are too complicated for analysis, we instead use empirical approximation. Let $m_a = \mathcal{N}(\mu_a, \sigma^2_a)$ be an incoming message (from the pre-activation variable to LeakyReLU). We assume that $\sigma^2_a = t$ is a constant and that $\mu_a \sim \mathcal{N}(0, 1)$ is a random variable. By sampling multiple means and then computing the outgoing messages (after applying LeakyReLU), we can approximate the average variance of the outgoing messages, as well as the average and empirical variance over means of the outgoing messages.

We computed these statistics for 101 different leak settings with 100 million samples each, and found that the relationship between leak and $\mu_{\mu_x}$ (average mean of the outgoing message) is approximately linear, while the relationships between leak and $\sigma^2_{\mu_x}$ or $\mu_{\sigma^2_x}$ are approximately quadratic. Using these samples, we fitted coefficients with an error margin below $5 \cdot 10^{-5}$. For our network, we chose a target variance of $1.5$ and a leak of $0.1$, resulting in

$$\sigma^2_x = 0.8040586726631379$$
$$\sigma^2_{\mu_x} \cdot (1 + \mu^2_{\mu_x}) = 0.44958619556324186.$$

These values are sufficient for now setting the prior variances of the inner linear layer according to the equations above. Finally, we set the prior variance of the biases to $0.5$, so that the output of each linear layer achieves an overall target prior predictive variance of approximately $t = 1.5 + 0.5 = 2.0$.

RESULTS IN PRACTICE

In practice, we found that the variance of the predictive posterior still goes up exponentially with the depth of the network despite our derived prior choices. However, if we lower the prior variance further to avoid this explosion, the network is overly restricted and unable to obtain a good fit during training. We therefore set the prior variances as outlined here, but acknowledge that choosing a good prior is still an unsolved problem.

# E  TABLES OF MESSAGE EQUATIONS

In the following, we provide tables summarizing all message equations used throughout our model. The tables are divided into three categories: linear algebra operations (Table 2), training signals (Table 3), and activation functions (Table 4). Each table contains the relevant forward and backward message equations, along with illustrations of the corresponding factor graph where necessary. These summaries serve as a reference for the mathematical operations performed during inference and training, and they will be valuable for factor graph modeling across various domains beyond neural networks.

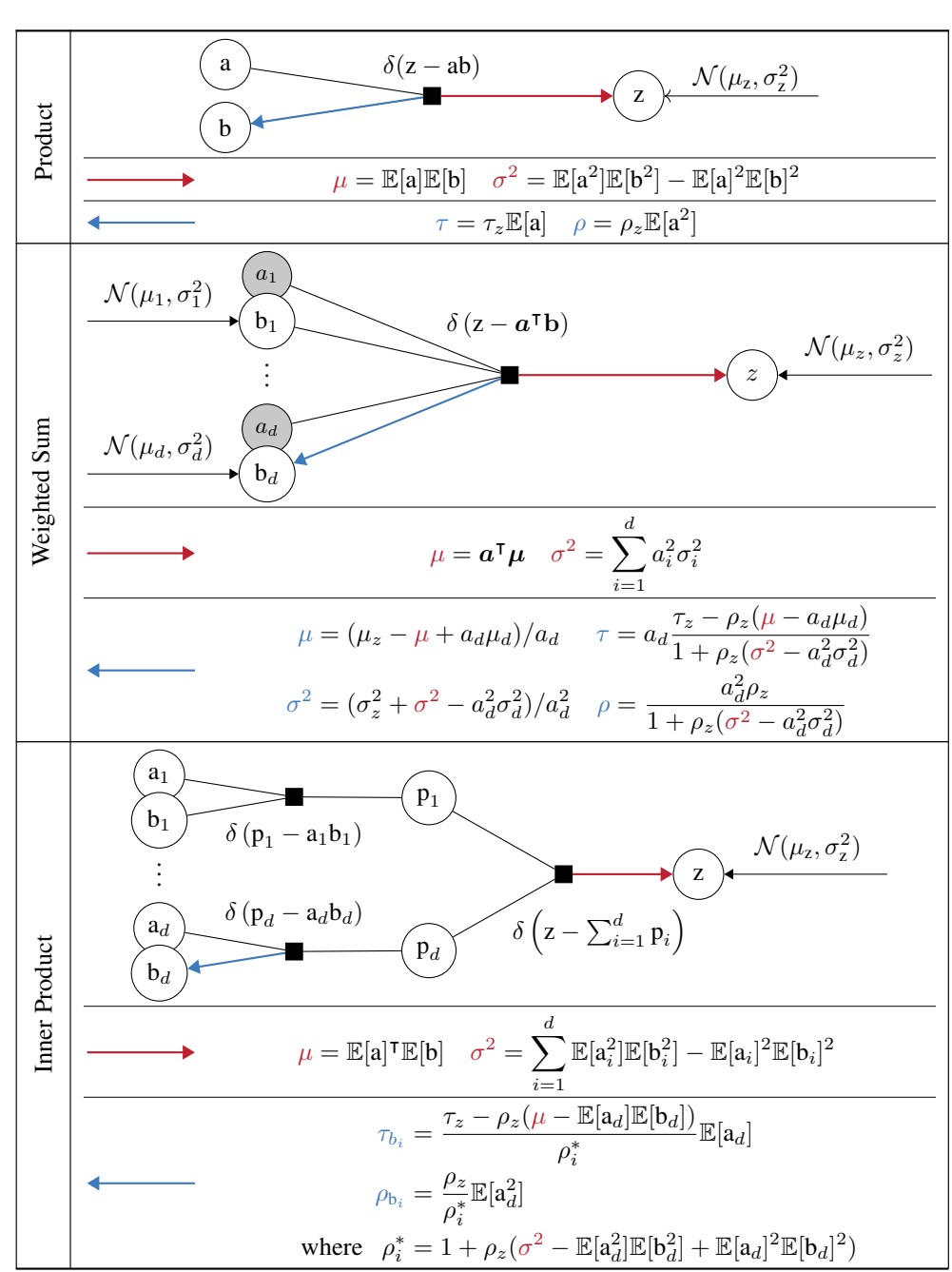

Table 2: Message equations for linear algebra: Calculating backward messages in natural parameters is preferable as it handles edge cases like $a_d = 0$ or $\rho_z = 0$ where location-scale equations are ill-defined. This approach also enhances numerical stability by avoiding division by very small quantities. Note that the inner product messages are simply compositions of the product and weighted sum messages with $a_i = 1$, $i = 1, \ldots, d$.

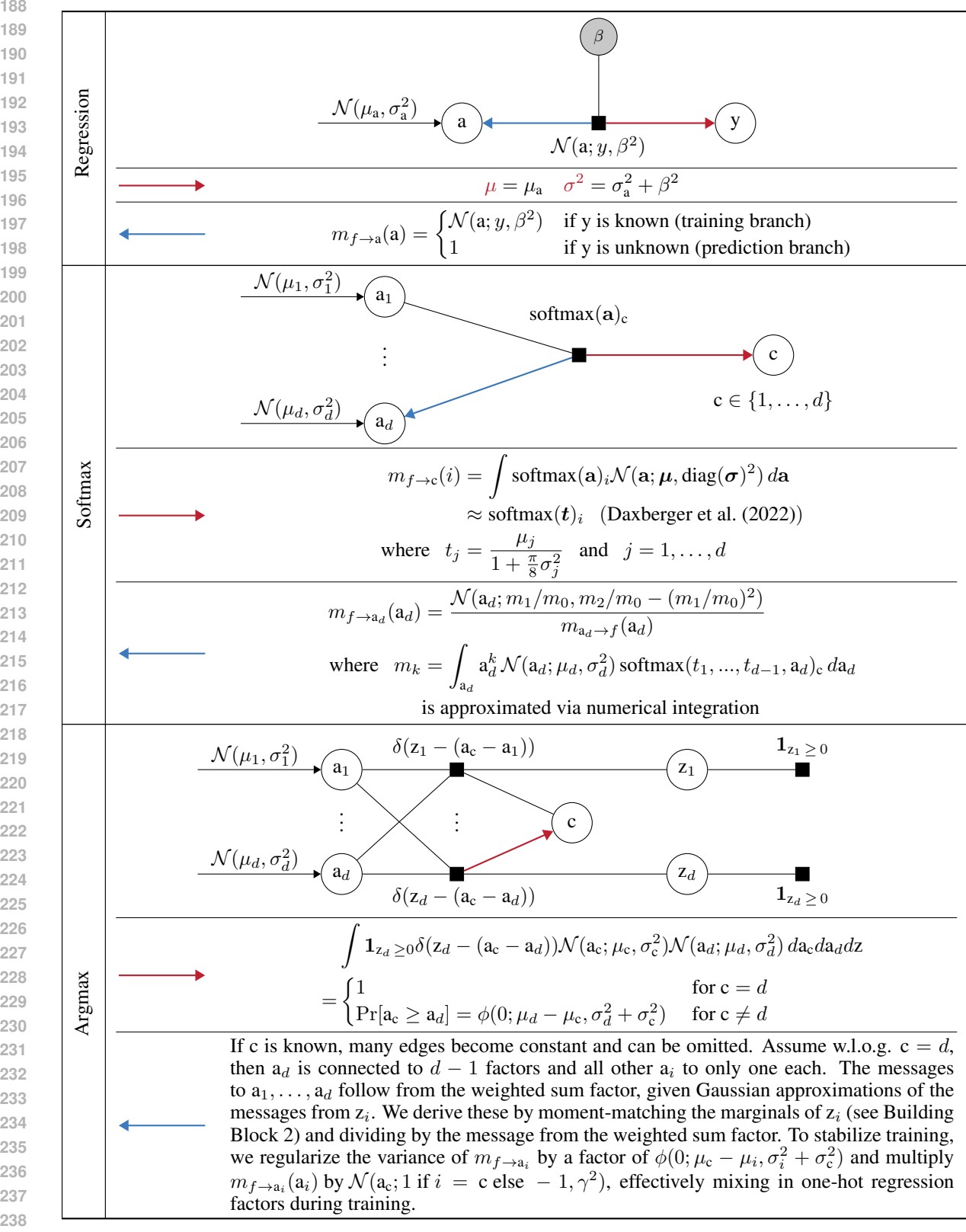

Table 3: Message equations for training signals. Note that the backward messages only apply in the case in which the target is known, i.e., on the training branches. On the prediction branch we only do foward passes.

| | |
|---|---|
| **Auxiliary Equations** | $\text{ReLUMoment}_k(\mu, \sigma^2) = \begin{cases} \mathbb{E}[\text{ReLU(a)}] \text{ with a} \sim \mathcal{N}(\mu, \sigma^2) & \text{for } k = 1 \\ \mathbb{E}[\text{ReLU}^2(\text{a})] \text{ with a} \sim \mathcal{N}(\mu, \sigma^2) & \text{for } k = 2 \end{cases}$ 

 $= \begin{cases} \sigma\varphi(x) + \mu\phi(x) & \text{for } k = 1 \\ \sigma\mu\varphi(x) + (\sigma^2 + \mu^2)\phi(x) & \text{for } k = 2 \end{cases}$ 

 where $\varphi$ and $\phi$ denote the pdf and cdf of $\mathcal{N}(0,1)$, respectively. 
<hr>
 $\zeta_k(\mu_1, \sigma_1, \mu_2, \sigma_2) := \int_0^\infty \text{a}^k \mathcal{N}(\text{a}; \mu_1, \sigma_1^2)\mathcal{N}(\text{a}; \mu_2, \sigma_2^2)\, d\text{a}$ 

 $= \mathcal{N}(\mu_1; \mu_2, \sigma_1^2 + \sigma_2^2) \cdot \begin{cases} \text{ReLUMoment}_k(\mu_m, \sigma_m^2) & \text{for } k = 1, 2 \\ \phi(\mu_m/\sigma_m) & \text{for } k = 0 \end{cases}$ 

 with $\tau_m = \dfrac{\mu_1}{\sigma_1^2} + \dfrac{\mu_2}{\sigma_2^2}, \quad \rho_m = \dfrac{1}{\sigma_1^2} + \dfrac{1}{\sigma_2^2}, \quad \mu_m = \dfrac{\tau_m}{\rho_m}, \quad \text{and } \sigma^2 = \dfrac{1}{\rho_m}$ 

 See Building Block 1 for the derivation of this equation. |
| **LeakyReLU** |  
<hr>
 We use marginal approximation while: 
 1. The outputs are finite and not NaN 
 2. Forward message: Precision of $m_{f\to z}$ is $\geq$ precision of $m_{a\to f}$, and $m_0 > 10^{-8}$ 
 3. Backward message: It has worked well to require $(\tau_z > 0) \vee (\rho_z > 2 \cdot 10^{-8})$ 
 Otherwise, we fall back to direct message approximation (forward) or $\mathbb{G}(0,0)$ (backward). |
| **Direct** | $\mu = (1 - \alpha) \cdot \text{ReLUMoment}_1(\mu_a, \sigma_a^2) + \alpha \cdot \mu_a$ 
 $\sigma^2 = (1 - \alpha^2) \cdot \text{ReLUMoment}_2(\mu_a, \sigma_a^2) + \alpha^2 \cdot (\sigma_a^2 + \mu_a^2) - \mu^2.$ |
| **Marginal** | $m_{f\to z}(\text{z}) = \dfrac{\mathcal{N}(\text{z}; \frac{m_1}{m_0}, \frac{m_2}{m_0} - (\frac{m_1}{m_0})^2)}{m_{z\to f}(\text{z})}$ 

 where $\quad m_k = (-1)^k \cdot \zeta_k(-\mu_a, \sigma_a^2, \ -\alpha \cdot \mu_z, \alpha^2 \cdot \sigma_z^2) + \zeta_k(\mu_a, \sigma_a^2, \ \mu_z, \sigma_z^2)$ 

 To compute the marginal backward message, set $\alpha_{\text{back}} = \alpha^{-1}$ 
 and swap $m_{a\to f}$ and $m_{z\to f}$ in the equation |

Table 4: Message equations for LeakyReLU with ReLU as the special case $\alpha = 0$.

