# OpenReview forum: "Scalable Approximate Message Passing for Bayesian Neural Networks"
_ICLR.cc/2025/Conference — Submitted to ICLR 2025_

### Official Review · Reviewer_yEQC · 2024-10-27

**Soundness:** 3
**Presentation:** 3
**Contribution:** 3
**Rating:** 6
**Confidence:** 3

**Summary:**

This paper advances message passing (MP) methods for Bayesian neural networks by introducing a novel framework that models the predictive posterior as a factor graph. The key technical contribution is being the first MP method to handle convolutional neural networks while avoiding double-counting of training data, which was a key limitation of previous approaches that led to overconfidence. Their method shows particularly strong performance in data-constrained settings, achieving 94.62% accuracy on MNIST with LeNet-5 using only 640 samples (compared to 22.15% for SGD), while also demonstrating better calibration and out-of-distribution detection than standard approaches. While the method scales to networks with 5.6 million parameters and requires minimal hyperparameter tuning, it currently has higher computational overhead compared to standard training methods and has not yet matched the scale of state-of-the-art variational inference approaches. Nevertheless, the work represents an important step forward in providing more balanced uncertainty estimates, especially in scenarios with limited training data.

**Strengths:**

- First message passing method to handle convolutional neural networks while solving the data double-counting problem
- Exceptional performance with limited data (94.62% MNIST accuracy with 640 samples vs 22.15% for SGD)
- Better calibration and out-of-distribution detection than standard approaches
- Scales to practical network sizes (5.6M parameters) with minimal hyperparameter tuning
- Clear theoretical framework with thorough derivations and implementation details, making it reproducible

**Weaknesses:**

- Significantly slower than standard training methods (96.4s vs 2.3s on GPU for LeNet-5 training)
- Does not yet match the scale of state-of-the-art variational inference methods
- Limited empirical evaluation - only tested on MNIST and synthetic data, lacking results on more complex datasets
- Memory intensive during training, and requiring approximately twice the memory of standard approaches during inference.
- No comparison against other Bayesian methods like variational inference or MCMC in terms of uncertainty quality
- Limited discussion of how the approach handles different neural network architectures beyond MLPs and basic CNNs
- Does not scale to large neural networks beyond 5.6M parameters.

**Questions:**

- Can you provide empirical results on more complex datasets beyond MNIST (e.g., CIFAR-10, ImageNet) to better demonstrate practical applicability and scalability?
- How does your method compare to modern variational inference approaches (like VOGN or IVON) in terms of uncertainty quality, calibration, and computational costs?
- What are the key bottlenecks causing the 40x slower training time compared to SGD, and are there potential optimizations to reduce this gap?
- How does your method handle modern neural network architectures with skip connections, batch normalization, or attention mechanisms - is the factor graph framework adaptable to these components?

---

> ### Author Response · Authors · 2024-11-28
> **Answers to Questions**
>
> Thank you for your review :) Your feedback has led to a major improvement of the paper in our view.
>
> The submitted revision contains answer to all four of your questions. In a nutshell:
>
> - In the revision we gave results on training a convnet of 890k parameters on CIFAR-10.
> - We compared the results against two of the strongest SOTA baselines available, namely AdamW and IVON each with a cosine annealing learning rate schedule.
> - In the conclusion section we are 100% transparent about major limitations, in particular the increased training time compared to AdamW. In a subsequent paragraph we address potential solutions to these issues.
> - Also in the conclusion we outline ideas on how to extend our framework to ResNets and transformers.
>
> While testing against stronger baseline methods than SGD we noticed that the superiority of our approach in data-constrained settings unfortunately vanishes. There we did not include data-constrained experiments on the CIFAR-10 dataset. This makes the MNIST experiments irrelevant, especially since it is a toy dataset anyways. We also removed the computational performance section. Instead we state in the limitations part of the conclusion that the training time, while scaling linearly in the model and dataset size, is typically on the order of one to two orders of magnitude higher than with AdamW using the highly optimized pytorch framework.

---

### Official Review · Reviewer_mv2H · 2024-10-30

**Soundness:** 3
**Presentation:** 2
**Contribution:** 3
**Rating:** 6
**Confidence:** 2

**Summary:**

This paper introduces a scalable message-passing (MP) framework for Bayesian neural networks (BNNs), aiming to improve uncertainty quantification in deep learning models. BNNs are known for their potential in high-stakes domains due to their ability to capture predictive uncertainty. Traditional methods, like variational inference (VI), struggle with overconfidence and hyperparameter sensitivity, prompting the development of this MP framework. The authors’ approach utilizes factor graphs to model the predictive posterior, demonstrating that this avoids common issues like double-counting training data, which leads to overconfidence in other MP methods. Their implementation shows better calibration and out-of-distribution detection than standard SGD, particularly in data-constrained settings.

**Strengths:**

* The MP approach achieves superior calibration and out-of-distribution detection, crucial for applications in high-stakes domains where understanding model uncertainty is vital.
* The method shows competitive accuracy, especially in scenarios with limited data, outperforming standard approaches like SGD on tasks with restricted data availability.
* This framework is the first to apply message passing effectively to convolutional neural networks, marking a significant advancement over previous MP implementations. It opens up a relatively unexplored branch which is very important for the advancement of the field.

**Weaknesses:**

* The MP method is computationally demanding, resulting in slower training times compared to standard approaches such as SGD and VI. The method needs substantial memory, especially during training, making it challenging to apply to memory-intensive tasks or very large networks without further optimization.
* Although it scales better than prior MP methods, the approach still lags behind VI in handling large, complex models and datasets.
* The method is based on some assumptions and oversimplification, such as the latent variables have a Gaussian distribution. This might not hold in complex scenarios with multimodal patterns.
* There is no proper time measurement of the method to have a crisp understanding of the computation complexity.

**Questions:**

* Would it be feasible to apply this approach to a larger network or a more challenging dataset?

---

> ### Author Response · Authors · 2024-11-28
> **Answer to Question**
>
> Thank you for your review! We appreciate that!
>
> We hope your question is answered in our reply to reviewer yEQC.

---

### Official Review · Reviewer_iRHR · 2024-11-03

**Soundness:** 3
**Presentation:** 3
**Contribution:** 2
**Rating:** 3
**Confidence:** 4

**Summary:**

They propose a scalable message-passing framework for Bayesian neural networks and derive message equations for various factors, which can benefit factor graph modeling across domains. The method is applied for both CNNs and FCNs on MNIST.

**Strengths:**

- Good contribution for the message passing community and especially the experiments on CNNs.

**Weaknesses:**

- The writeup in overall is difficult to follow and the motivation of the work is not clear.
- Poor experimental evaluation, especially since the method is tested only against synthetic data and MNIST. It should be tested on bigger models than LeNet (which is quite outdated), and it should also be tested against CIFAR-10 which is a common benchmark on BNNs.
- All of the proofs are in the Appendix. I would suggest to squeeze something in the main text since there is still space in page 10. Probably the main proof of global minimization objective?

**Questions:**

My main question is why have you not evaluated the proposed approach on CIFAR10 and you evaluated only against MNIST? MNIST is considered as a toy dataset so we do not know if your method generalizes in bigger models even though you scaled the MLP model on MNIST. In the current state the paper is not strongly either theoretically or experimental and therefore it needs more revisions.


Misc: line 178 "predictionsfor" -> "predictions for"

---

> ### Author Response · Authors · 2024-11-19
> **Evaluation on CIFAR-10 (preliminary)**
>
> Thank you for your review!
>
> We now have trained a convnet with our approach, using roughly 900k parameters on CIFAR-10, and achieved over 77% validation accuracy after 25 epochs (with no signs of saturation). To provide perspective, we trained the same model using Torch with the Adam optimizer (with default parameters—which, after a hyperparameter search, turned out to be quite optimal) and achieved a similar validation accuracy of around 75% after 25 epochs. Of course, the Torch library is highly optimized, and our method inherently has an overhead compared to training a deterministic network. So when giving the two methods the same compute budget, Torch + Adam would almost certainly achieve better accuracy. We do not claim otherwise, and that is not the point. Rather, we want to demonstrate that it is possible to train convolutional BNNs with a gradient-free message-passing algorithm while avoiding the double counting problem.
>
> The reason we did not do this prior to submission was that, in the implementation state of the main branch of the repository, the GPU memory footprint exceeded 80 GB when training such a model on CIFAR-10. Prior to submission, we did not have time to implement our ideas on how to reduce the GPU memory footprint. Now we have implemented one of our ideas—namely, frozen batch buffering. Instead of storing the weight aggregates of inactive batches, which we dub frozen batches, entirely on the GPU, we now store them in main memory and buffer them into GPU memory once they are needed. This made it possible to train the above-mentioned model on a single GPU.
>
> In the revised version of the paper, we plan to:
> a) provide a more thorough evaluation on CIFAR-10;
> b) explain in detail how the memory requirements scale with parameters such as batch and model size; and
> c) give ideas for future work on how to decrease training time and memory consumption.
> Probably, this will not leave as any space to squeeze in proofs from the appendix into the main text.
>
> Additionally, we would appreciate your feedback on which areas of the paper are particularly difficult to follow. This will help us focus our efforts.
>
> Related concern: Considering that state-of-the-art models now have literally trillions of parameters and are trained on petabytes of data, do you think the term "scalable" in the paper's title is fitting with modern standards?

---

> > ### Comment · Reviewer_iRHR · 2024-11-22
> >
> > Hello authors and thank you for your reply. I have read your reply, and it looks like your method is indeed not scalable. That being said I think that the paper is not very strong in its current form. Since the method is not scalable to large datasets, then it should be focused more on theoretical advances such as some kind of proof of convergence.

---

> > > ### Author Response · Authors · 2024-11-22
> > >
> > > Thanks for your reply!
> > >
> > > One point we would like to stress: the difference of runtime on GPU of our approach compared to non-Bayesian optimized PyTorch implementation is only a factor of 6 (and PyTorch is highly optimized over the past few years) while the message passing algorithm for Bayesian neural networks offers the advantage of calibrated model uncertainties. This permits both to learn highly accurate predictive models on a subset of training data *and* increase the accuracy in prediction much faster when rejecting test examples based on these calibrated probabilities.
> > >
> > > We would also like to note that our submission is one of the first - maybe *the* first - that demonstrates how to scale the training of calibrated Bayesian Neural networks with millions of weights and trained on tens of thousands of training examples. This is an algorithmic advancement over variational inference-based learning of Bayesian neural networks (which also overfit more than our inference technique).

---

> > > > ### Comment · Reviewer_iRHR · 2024-11-22
> > > >
> > > > Yes I am well aware of that and from the current literature. Message parsing algorithms are not very famous or "trending" lately so it's good to see some kind of alternative approaches to BNNs. My main concern is still that the method is not very scalable. For example most new works should have something involving at least a Res-Net18 if they are experimental papers. And since your work is not scalable for that I would highly suggest to submit it with more theoretical findings.

---

### Meta-Review · Area_Chair_JxEN · 2024-12-20

**Metareview:**

The paper develops approximate message passing as an approximate inference technique for Bayesian neural networks. The procedure is fairly involved and requires representing the neural network as a factor graph. Then, the authors perform loopy belief propagation with approximations to the message passing steps. The final results is a diagonal Gaussian approximation to the posterior. The authors show that the method is competitive with variational inference-based methods on a synthetic dataset with MLP and on CIFAR-10 with a small convolutional model.

Strengths:
- The main strength of the paper is in developing novel approximate message passing methodology applicable to (small) neural network models.
- The authors develop and describe a number of tricks and approximations to get the method to run and address numerical instabilities.
- The method is competitive with variational inference in small-scale experiments.

Weaknesses:
- The proposed method is involved, and it is not clear what are the advantages over existing Bayesian methods for neural networks, even variational inference.
- The authors do not do a careful literature review of existing Bayesian methods and do not compare to relevant baselines.
  + Stochastic gradient Monte Carlo methods are popular, scalable, and avoid many limitations of VI [1, 2].
  + For the scale of experiments in the paper, full HMC is easily applicable [3]
  + Deep ensembles should be treated as a relevant baseline for uncertainty calibration [4]
  + In the rebuttal, the authors say:
  > We would also like to note that our submission is one of the first - maybe the first - that demonstrates how to scale the training of calibrated Bayesian Neural networks with millions of weights and trained on tens of thousands of training examples.
    * SWAG [5] proposed in 2019 was run on ImageNet (1M+ datapoints, 1k classes) with ResNet-152 models.
  +  Other scalable Bayesian methods include Laplace approximation-based methods, MC-dropout, etc.
- The empirical evaluation is not sufficient for confirming that the proposed method provides any advantages over baselines.
  + There are only two experiments: synthetic dataset and CIFAR-10. For CIFAR-10, the accuracy is 78% for the best method. The state-of-the-art accuracy on CIFAR-10 is 99+% for pretrained models and 97+% for models trained from scratch. While the authors use a small model, 78% is so far from the state-of-the-art that this result cannot be used to argue for the method providing a practical improvement.
  + Moreover, in the experiments, the proposed MP method loses to the AdamW baseline on accuracy.
  + While the method provides improved calibration, I don't believe that is the correct target for Bayesian methods. Calibration is a highly-sensitive metric that can be improved with simple interventions [6]. I believe Bayesian methods should aim to improve accuracy, or provide some other benefit over the baselines.
- The method is computationally expensive compared to baselines, at least in the current implementation. The current implementation also does not support normalization layers and residual connections.

Decision recommendation: The paper makes a meaningful contribution in developing an approximate message passing algorithm for Bayesian neural network. In its current form, it is a proof of concept, as the method is not applicable to modern models and does not provide significant improvements in the experiments. The empirical evaluation is very limited and missing relevant baselines, but even compared to the presented baselines the method does not make a significant improvement. Thus, I recommend rejecting the paper in its current form, and encourage the authors to improve the empirical evaluation.

[1] Bayesian Learning via Stochastic Gradient Langevin Dynamics; Max Welling, Yee Whye Teh

[2] Stochastic Gradient Hamiltonian Monte Carlo; Tianqi Chen, Emily B. Fox, Carlos Guestrin

[3] What Are Bayesian Neural Network Posteriors Really Like?; Pavel Izmailov, Sharad Vikram, Matthew D Hoffman, Andrew Gordon Gordon Wilson

[4] Simple and Scalable Predictive Uncertainty Estimation using Deep Ensembles; Balaji Lakshminarayanan, Alexander Pritzel, Charles Blundell

[5] A Simple Baseline for Bayesian Uncertainty in Deep Learning; Wesley J. Maddox, Pavel Izmailov, Timur Garipov, Dmitry P. Vetrov, Andrew Gordon Wilson

[6] On Calibration of Modern Neural Networks; Chuan Guo, Geoff Pleiss, Yu Sun, Kilian Q. Weinberger

**Additional Comments On Reviewer Discussion:**

The reviews for the paper were mixed: 6,6,3. The reviewers raised concerns with scalability of the proposed method, and the limited evaluation. The authors responded with new experiments on CIFAR-10. Notably, the reviewers were initially impressed with results in a data limited setting on MNIST. However, the authors removed those experiments completely from the paper during the rebuttal phase:
> While testing against stronger baseline methods than SGD we noticed that the superiority of our approach in data-constrained settings unfortunately vanishes. There we did not include data-constrained experiments on the CIFAR-10 dataset. This makes the MNIST experiments irrelevant, especially since it is a toy dataset anyways.

---

### Decision · Program_Chairs · 2025-01-22

Reject